EMBO
Molecular Medicine

# A circRNA signature predicts postoperative recurrence in stage II/III colon cancer

Huai-Qiang Ju[1,†] (iD), Qi Zhao[1,†] (iD), Feng Wang[1,2,†], Ping Lan[3,†], Zixian Wang[1,2,†], Zhi-Xiang Zuo[1,4], Qi-Nian Wu[1,5], Xin-Juan Fan[3], Hai-Yu Mo[1], Li Chen[4], Ting Li[1], Chao Ren[1], Xiang-Bo Wan[3], Gong Chen[1,6], Yu-Hong Li[1,2], Wei-Hua Jia[1] & Rui-Hua Xu[1,2,*] (iD)

## Abstract

Accurate risk stratification for patients with stage II/III colon cancer is pivotal for postoperative treatment decisions. Here, we aimed to identify and validate a circRNA-based signature that could improve postoperative prognostic stratification for these patients. In current retrospective analysis, we included 667 patients with R0 resected stage II/III colon cancer. Using RNA-seq analysis of 20 paired frozen tissues collected postoperation, we profiled differential circRNA expression between patients with and without recurrence, followed by quantitative validation. With clinical information, we generated a four-circRNA-based cirScore to classify patients into high-risk and low-risk groups in the training cohort. The patients with high cirScores in the training cohort had a shorter disease-free survival (DFS) and overall survival (OS) than patients with low cirScores. The prognostic capacity of the classifier was validated in the internal and external cohorts. Loss-of-function assays indicated that the selected circRNAs played functional roles in colon cancer progression. Overall, our four-circRNA-based classifier is a reliable prognostic tool for postoperative disease recurrence in patients with stage II/III colon cancer.

**Keywords** biomarker; circular RNA; recurrence; stage II/III colon cancer
**Subject Categories** Biomarkers; Cancer

## Introduction

Approximately 60% of patients with colon cancer present with stage II/III disease (Rabeneck *et al*, 2015). Surgical resection is the only possible cure for these patients (Rabeneck *et al*, 2015). However, there are still 20–30% of patients who suffer from postoperative recurrence, which results in dismal survival (O'Connell *et al*, 2008; Andre *et al*, 2009). Traditionally, adjuvant chemotherapy has been the standard of care for patients with high-risk stage II, defined by clinicopathological features such as T4 lesion and the retrieval of < 12 lymph nodes, and all stage III colon cancer, defined as N1/N2M0 disease irrespective of T stage. However, clinicopathological risk factors and microsatellite instability status do not adequately distinguish between patients who have a high or low risk of disease recurrence, thereby not indicating which patients are likely to benefit from postoperative chemotherapy (Gray *et al*, 2007; Morris *et al*, 2007). In view of this clinical challenge, there is an unmet need for novel recurrence-specific molecular biomarkers that allow for better prognostic stratification and more appropriate therapies for patients with stage II/III colon cancer.

Circular RNA (circRNA), a rediscovered, abundant RNA species, is a type of non-coding covalent closed RNAs formed from both exonic and intronic sequences (Morris & Mattick, 2014; Chen & Yang, 2015). circRNAs are characterized by several properties, such as being evolutionarily conserved, having tissue-specific expression, more stable than linear miRNA (Jeck *et al*, 2013; Memczak *et al*, 2013; Taborda *et al*, 2017). They can regulate gene expression, acting as real sponges for miRNAs, and also regulate alternative splicing or act as transcriptional factors and inclusive encoding for proteins (Taborda *et al*, 2017). However, to the best of our knowledge, the ability of circRNA-based signatures as novel prognostic biomarkers for colon cancer has not yet been comprehensively investigated.

In this study, we conducted a multicenter, retrospective study to assess the ability of circRNA expression profiles to predict disease recurrence in patients with stage II/III colon cancer. We aimed to identify and validate a circRNA-based signature that could improve postoperative prognostic stratification for these patients.

1 State Key Laboratory of Oncology in South China, Collaborative Innovation Center for Cancer Medicine, Sun Yat-sen University Cancer Center, Guangzhou, China
2 Department of Medical Oncology, Sun Yat-sen University Cancer Center, Guangzhou, China
3 The Sixth Affiliated Hospital, Sun Yat-sen University, Guangzhou, China
4 State Key Laboratory of Biocontrol, School of Life Sciences, Sun Yat-sen University, Guangzhou, China
5 Department of Pathology, Sun Yat-sen University Cancer Center, Guangzhou, China
6 Department of Colorectal Surgery, Sun Yat-sen University Cancer Center, Guangzhou, China
*Corresponding author. Tel: +86 20 8734 3333; E-mail: xurh@sysucc.org.cn
†These authors contributed equally to this work

# Results

## Clinicopathological features of patients

As shown in Fig 1, the frozen tissue samples of 667 colon cancer patients with stage II/III disease were collected from Sun Yat-sen University Cancer Center (487 samples) for discovery ($n = 20$), selection ($n = 96$), training ($n = 249$), and internal validation ($n = 122$), and the Six Affiliated Hospital of Sun Yat-sen University for external validation ($n = 180$). The detailed clinicopathological characteristics of the training, and the internal and external validation datasets are shown in Table 1. All patients had undergone surgical resection with histologically negative resection margins. The median follow-up periods were 67 months (IQR, 50–78), 66 months (IQR, 48–79), and 57 months (IQR, 48–64), respectively, in the training and internal and external validation sets. The corresponding 5-year disease-free survival (DFS, defined as the time from the date of surgery to the date of confirmed tumor relapse or death from any cause, as the outcome) rates were 72.6% (95% CI, 68.1–77.3), 69.5% (95% CI, 61.6–78.4), and 75.2% (95% CI, 69.3–81.6), and 5-year overall survival (OS, defined as the time from the date of surgery to the date of death or the last known follow-up) rates were 81.1% (95% CI, 77.1–85.3), 78.1% (95% CI, 70.8–86.1), and 82.4% (95% CI, 77.1–88.0).

## Selection and validation of candidate circRNAs

Based on the RNA-seq data and bioinformatics analysis, differential expression analysis identified 437 circRNAs (326 upregulated and 111 downregulated, marked as "TNcircles" afterward) between the tumor and adjacent normal tissues by using a soft threshold. The analysis also identified 103 differentially expressed circRNAs (48 upregulated and 55 downregulated, marked as "RNcircles" afterward) between recurrent and non-recurrent tumor tissues. Both TNcircles and RNcircles showed strong classification properties in distinguishing each of the groups (Fig 2A and B). In addition, the differential expression results indicated that circRNAs experienced more prominent changes between the normal and tumor tissues than between the recurrent and non-recurrent tumor tissues (Fig 2A and B).

Next, we investigated whether circRNAs could be used as prognostic biomarkers in patients with stage II/III colon cancer. First, 38 significantly upregulated circRNAs were selected from TNcircles for further validation according to the aforementioned retaining criteria. In addition, we prioritized 62 circRNAs from RNcircles using the same selection criteria to obtain a total of 100 circRNAs for validation assays. Considering a potential false discovery that might be introduced by the inadequate sensitivity of the RNA-seq and sample size, we enrolled 48 recurrent and 48 non-recurrent samples for further validation using qRT–PCR assay. Among these candidates, 22 circRNAs (10 from TNcircles and 12 from RNcircles) were further selected based on the extremely significant difference ($P < 0.01$; Figs 2C and EV1). We quantified these 22 circRNAs with qRT–PCR in the training cohort ($n = 249$) and further reduced the number of candidates using the (least absolute shrinkage and selection operator) LASSO-bagging procedure as described in Materials and Methods (Fig 2D). Finally, we obtained four circRNAs that were strongly

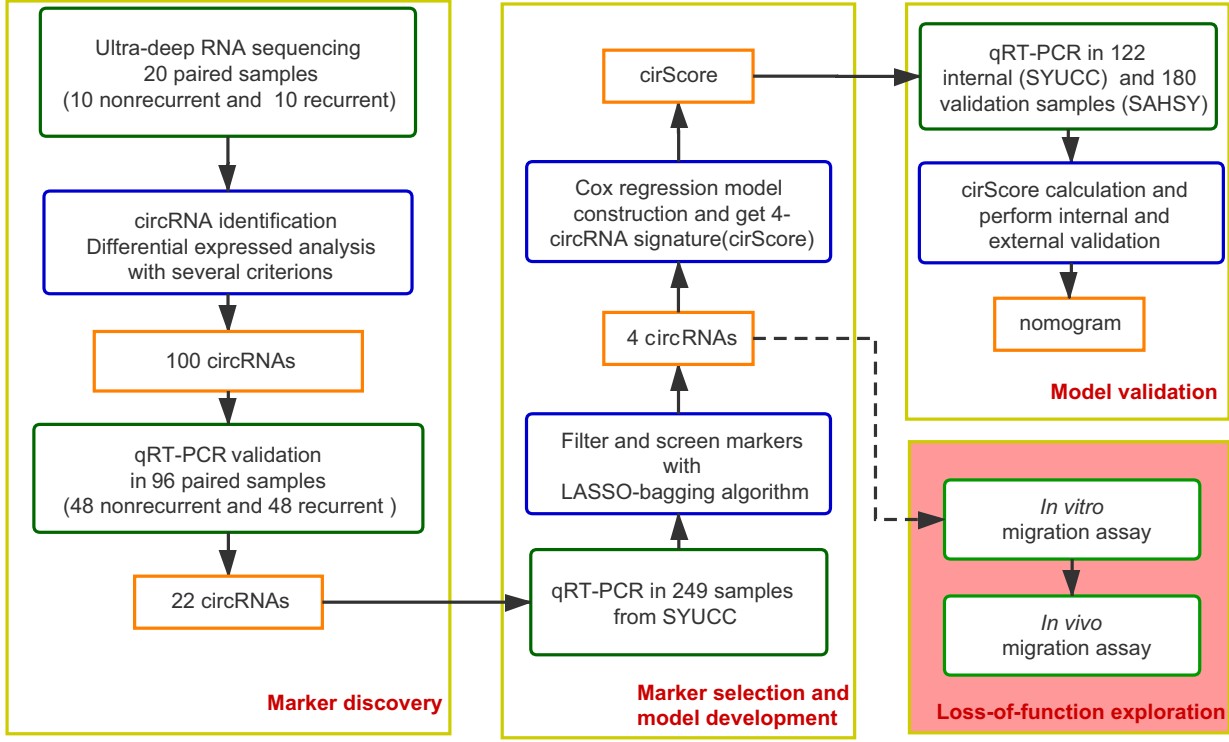

**Figure 1. Study flowchart.**

SYUCC = Sun Yat-sen University Cancer Center. SAHSY = the Six Affiliated Hospital of Sun Yat-sen University. Training and internal validation sets were randomly selected at a 2:1 ratio from the samples from SYUCC.

**Table 1. Clinical characteristic of patients with stage II/III colon cancer involved in this study.**

| | Training set (n = 249) | Internal validation set (n = 122) | External validation set (n = 180) |
|---|---|---|---|
| Age | | | |
| ≥65 year | 168 (67.5) | 84 (68.9) | 146 (81.1) |
| < 65 year | 81 (32.5) | 38 (31.1) | 34 (18.9) |
| Sex | | | |
| Male | 106 (42.6) | 54 (44.3) | 64 (35.6) |
| Female | 143 (57.4) | 68 (55.7) | 116 (64.4) |
| Primary tumor location | | | |
| Left-sided | 180 (72.3) | 91 (74.6) | 43 (23.9) |
| Right-sided | 69 (27.7) | 31 (25.4) | 137 (76.1) |
| Perineural invasion | | | |
| Yes | 177 (71.1) | 85 (69.7) | 138 (76.7) |
| No | 72 (28.9) | 37 (30.3) | 42 (23.3) |
| Lymphatic or vascular invasion | | | |
| Yes | 204 (81.9) | 94 (77) | 161 (89.4) |
| No | 45 (18.1) | 28 (23) | 19 (10.6) |
| Tumor differentiation | | | |
| Well or moderately differentiated | 176 (70.7) | 89 (73) | 136 (75.6) |
| Poorly differentiated or undifferentiated | 73 (29.3) | 33 (27) | 44 (24.4) |
| Mismatch repair status | | | |
| Mismatch repair-deficient | 27 (10.8) | 6 (4.9) | 19 (10.6) |
| Mismatch repair-proficient | 76 (30.5) | 37 (30.3) | 64 (35.5) |
| Unexamined | 146 (58.6) | 79 (64.8) | 97 (53.9) |
| T stage | | | |
| T1 | 2 (0.8) | 2 (1.6) | 1 (0.6) |
| T2 | 7 (2.8) | 3 (2.5) | 8 (4.4) |
| T3 | 118 (47.4) | 51 (41.8) | 147 (81.7) |
| T4 | 122 (49) | 66 (54.1) | 24 (13.3) |
| N stage | | | |
| N0 | 128 (51.4) | 57 (46.7) | 77 (42.8) |
| N1 | 73 (29.3) | 41 (33.6) | 77 (42.8) |
| N2 | 48 (19.3) | 24 (19.7) | 26 (14.4) |
| The total evaluated lymph node count | | | |
| < 12 | 79 (31.7) | 41 (33.6) | 5 (2.8) |
| ≥ 12 | 170 (68.3) | 81 (66.4) | 175 (97.2) |
| Clinical risk group[a] | | | |
| Non-high-risk stage II | 27 (10.8) | 6 (4.9) | 27 (15) |
| High-risk stage II | 91 (36.5) | 49 (40.2) | 39 (21.7) |
| Non-high-risk stage III | 43 (17.3) | 15 (12.3) | 81 (45) |
| High-risk stage III | 88 (35.3) | 52 (42.6) | 33 (18.3) |

Data are n (%).

[a]Stage II disease was considered high-risk if positive for the biomarkers for poorly differentiated or undifferentiated histology (exclusive of mismatch repair-deficient cases), perineural invasion, lymphatic or vascular invasion, or T4 stage II. Stage III disease was considered high-risk if it was staged T4, N2, or both.

predictive of DFS, i.e., hsa_circ_0122319, hsa_circ_0087391, hsa_circ_0079480, and hsa_circ_0008039 (Fig 2D). Notably, multivariate Cox regression analysis showed that these four circRNAs are mutually independent (Appendix Table S1). We also observed that the predicting performance of the four-circRNA-based risk score (cirScore) mostly outperforms than single circRNA with the time-dependent AUC analysis (Fig 2E). The circularity and stability of the four selected circRNAs were verified by Sanger sequencing and RNase R treatment. After examined by RT–PCR with divergent primers, the sequenced PCR product was corresponding from the bioinformatics analysis with the exact back-splice junction (Fig EV2A). We next validated the circularity of these candidates by RNase R treatment, and the mouse GAPDH mRNA was used as spike-in for normalization. The results indicated that these circRNAs were more resistance to digestion with RNase R exonuclease compared with linear host genes, which further confirmed that these circRNAs harbors a circular RNA structure (Fig EV2B). Taken together, these results indicated that the circRNA may be served as novel prognostic biomarkers for colon cancer.

## Construction and validation of the four-circRNA-based prognostic model

Then, a risk score was calculated for each patient using a formula derived from the expression levels of the four circRNAs weighted by their regression coefficient:

$$cirScore = 0.46 \times \mathrm{Exp}_{hsa\_circ\_0122319} + (-0.386 \times \mathrm{Exp}_{hsa\_circ\_0083791}) + 0.293 \times \mathrm{Exp}_{hsa\_circ\_0079480} + 0.439 \times \mathrm{Exp}_{hsa\_circ\_0008039}$$

Using the cirScore, we divided patients into high- and low-risk groups with its median value ($-0.323$) among the training cohort. Survival analysis showed that patients in the high-risk group had a poorer DFS than those in the low-risk group (hazard ratio [HR], 4.38; 95% confidence interval [CI], 2.52–7.64, $P < 0.0001$; Fig 3A). Moreover, we observed a similar impact of the cirScore on OS (high $vs$. low risk, HR, 5.13, 95% CI, 2.56–10.16, $P < 0.001$; Fig 3B).

To validate the prognostic prediction performance of the cirScore, patients in the internal and external validation cohorts were classified into high- and low-risk groups using the same cutoff obtained from the training cohort. In the internal validation cohort, patients with a high cirScore had a shorter DFS (HR, 2.89, 95% CI, 1.37–6.09, $P < 0.001$; Fig 3C) and a shorter OS (HR, 4.22, 95% CI, 1.61–11.03, $P < 0.001$; Fig 3D). Likewise, a high cirScore was associated with worse DFS (HR, 3.63, 95% CI, 1.81–7.29, $P < 0.01$; Fig 3E) and OS (HR, 4.25, 95% CI, 1.9–9.54, $P < 0.0001$; Fig 3F) in the external validation cohort.

After adjustment for baseline clinicopathologic factors, the cirScore remained a powerful and significant predictor of DFS and OS in the training set (HR = 4.64 [95% CI, 2.64–8.17], $P < 0.0001$ and HR = 5.45 [95% CI, 2.70–11.00], $P < 0.0001$, respectively). We also noted similar results in the internal validation set (HR = 2.96 [95% CI, 1.37–6.42], $P = 0.0058$ for DFS and HR = 3.82 [95% CI, 1.44–10.15], $P = 0.007$ for OS) and in the external validation set (HR = 2.50 [95% CI, 1.16–5.36], $P = 0.008$ for DFS and HR = 4.15 [95% CI, 1.79–9.64], $P = 0.0009$ for OS).

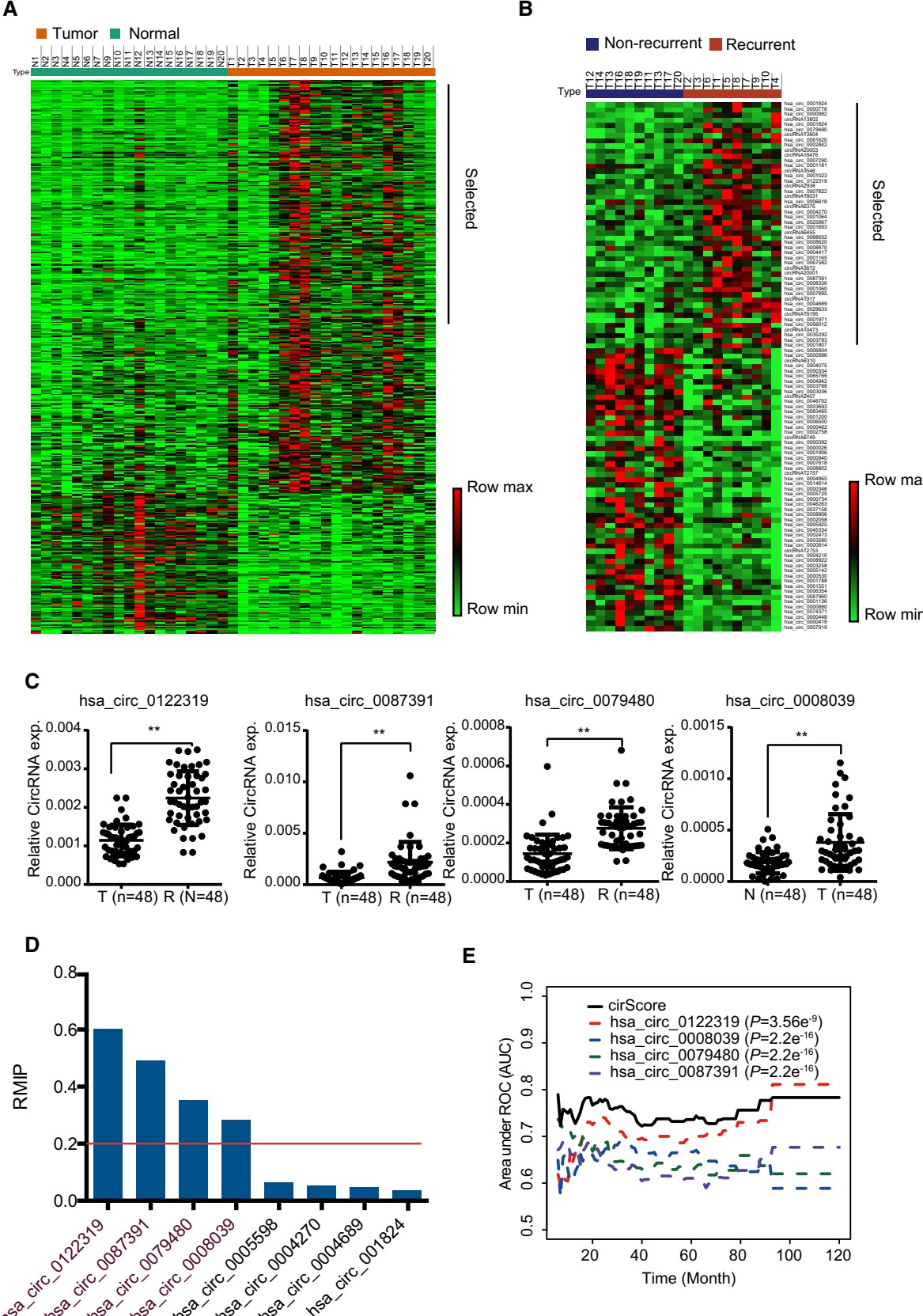

**Figure 2.**

◀

**Figure 2. Marker validation and selection from the circRNA-sequencing experiment.**

A  Expression profiling of differentially expressed circRNAs between the tumor and normal groups. Rows represent circRNAs, and columns represent samples. Rows were ordered by fold change, and columns were ordered by their group. The sample of N8 was not included due to low sequencing library size.

B  Expression profiling of differentially expressed circRNAs between the recurrence and non-recurrence groups. Both the row and column were unsupervised and clustered with the hierarchical clustering method.

C  The 4 of 22 differentially expressed circRNAs were confirmed by qRT–PCR, which were retained after marker selection procedure. **$P < 0.01$, Student's $t$-test, mean ± SD.

D  Bar plot shows the resample model inclusion proportion (RMIP) of qualified circRNAs calculated in the training dataset. The red line presents the threshold used to obtain the final markers.

E  Time-dependent AUC analysis of individual circle RNA and cirScore for predicting recurrence in the training dataset. $P$-values are shown for the indicated comparison of AUC between each marker and cirScore. Student's $t$-test, AUC = area under the curve.

Data information: Exact $P$-values are specified in Appendix Table S5.
Source data are available online for this figure.

## Loss-of-function assay of selected circRNAs regulating cell metastasis

We thus determined to evaluate the biological roles of the selected circRNA in colon cancer. Among the four circRNA markers, three circRNA (hsa_circ_0122319, hsa_circ_0079480, and hsa_circ_0087391) were significantly overexpressed in the recurrent samples and in the colon cancer cells (Figs 2C and EV3A). The circularity of these circRNAs was further verified by RT–PCR with divergent or convergent primers (Figs 4A and EV3B). To assess whether these circRNAs promoted colon cancer progression, SW620 and HCT116 cells with high metastatic potential were used to conduct loss-of-function assay by lentivirus-mediated stable gene silencing. The knockdown efficiency and specificity were verified by qRT–PCR, immunoblotting, and RNA-seq analysis. The results demonstrated that knockdown of these circRNAs had no effects on the mRNA or protein expression of the host genes (Figs 4B and EV3C and D), and had a high similarity of gene expression profile between two independent shRNA group in SW620 and HCT116 cells (Fig EV3E), suggesting that the following regulatory effects directly result from targeting the circRNAs rather than off-targets. Remarkably, knockdown of these circRNAs using two independent shRNAs significantly suppressed cell migration capacity in the detected cells (Fig 4C and D).

Subsequently, to further determine the oncogenic effects of representative circRNA in promoting colon cancer metastasis *in vivo*, the hsa_circ_0079480 knockdown and control cells were injected into the distal tip of the mice spleen using a Hamilton syringe. Six weeks later, the mice were sacrificed, and the spleen and liver were removed and embedded in paraffin. All the mice ($N = 8$ per group) had tumors that formed in the spleen. Moreover, the number of metastatic nodules in the livers was significantly reduced in mice injected with hsa_circ_0079480 knockdown cells compared with those injected with control colon cancer cells (Fig 4E and F). We further explored the role of hsa_circ_0079480 in lung colonization by injecting colon cancer cells directly into the tail veins of nude mice ($N = 8$ per group). The mice injected with control colon cancer cells induced a heavy lung metastatic burden as verified by histologic examination, whereas knockdown of hsa_circ_0079480 almost abolished lung metastasis (Fig 4G and H). The loss-of-function assay indicated that the circRNAs might play functional roles in the sophisticated regulation of colon cancer progression.

## Stratified analysis with known risk factors

We further performed stratified survival analyses to assess the prognostic performance of the cirScore against the clinical risk-stratification scheme (i.e., the high- and low-risk stage II and high- and low-risk stage III groups). Stage II disease was considered high-risk if it was presented with poorly differentiated or undifferentiated histology (exclusive of mismatch repair-deficient cases), perineural invasion, lymphatic or vascular invasion, or T4 status. Stage III disease was considered high-risk if it was staged T4, N2, or both. All three study cohorts were combined to obtain an increased statistical power for stratified survival analyses. As a result, in both the low- and high-risk stage II group, patients with high cirScore had a shorter DFS (HR = 7.72 [95% CI, 0.9–66.44, $P = 0.028$ and HR = 2.03 [95% CI, 1.06–3.90], $P = 0.0290$; Fig EV4A] than those with low cirScore. Additionally, in both the non-high-risk and high-risk stage III groups, patients were further stratified by the cirScore into subgroups with significantly different DFS (high *vs.* low cirScore: HR = 3.37 [95% CI, 1.90–15.97], $P < 0.0001$ and HR = 7.62 [95% CI, 3.16–18.41], $P < 0.0001$, respectively; Fig EV4A). Moreover, similar findings were obtained regarding the impact of the cirScore on OS after stratified by the clinical risk-stratification scheme (Fig EV4B). To note, result from the low-risk stage II did not reach the statistical significance ($P = 0.21$; Fig EV4C). Time-dependent receiver operating characteristic (ROC) analyses revealed that the combination of the cirScore with the clinical risk-stratification scheme achieved a superior prognostic accuracy to the clinical risk-stratification scheme alone for DFS and OS in the training set, and the internal and external validation sets (Fig EV4C).

## Building nomograms and time-dependent ROC analysis

Through a stepwise backward selection process on the basis of AIC, the cirScore, age at diagnosis, N stage, NI, and VI remained in the final Cox model for DFS (Appendix Table S2). To develop a clinically applicable tools that could provide individualized estimation of the 3- or 5-year DFS, a nomogram was established based on the final Cox model for DFS (Fig 5A). The nomogram achieved a C-index of 0.816 (95% CI, 0.774–0.857), and the calibration plots showed close agreement between the actual DFS probabilities and the predicted DFS from the nomogram in the training set (Fig 5B). The C-indices were 0.789 (95% CI, 0.719–0.859) and 0.694 (95% CI, 0.608–0.780), respectively, in the internal and external validation sets. The actual DFS probabilities were consistent with the

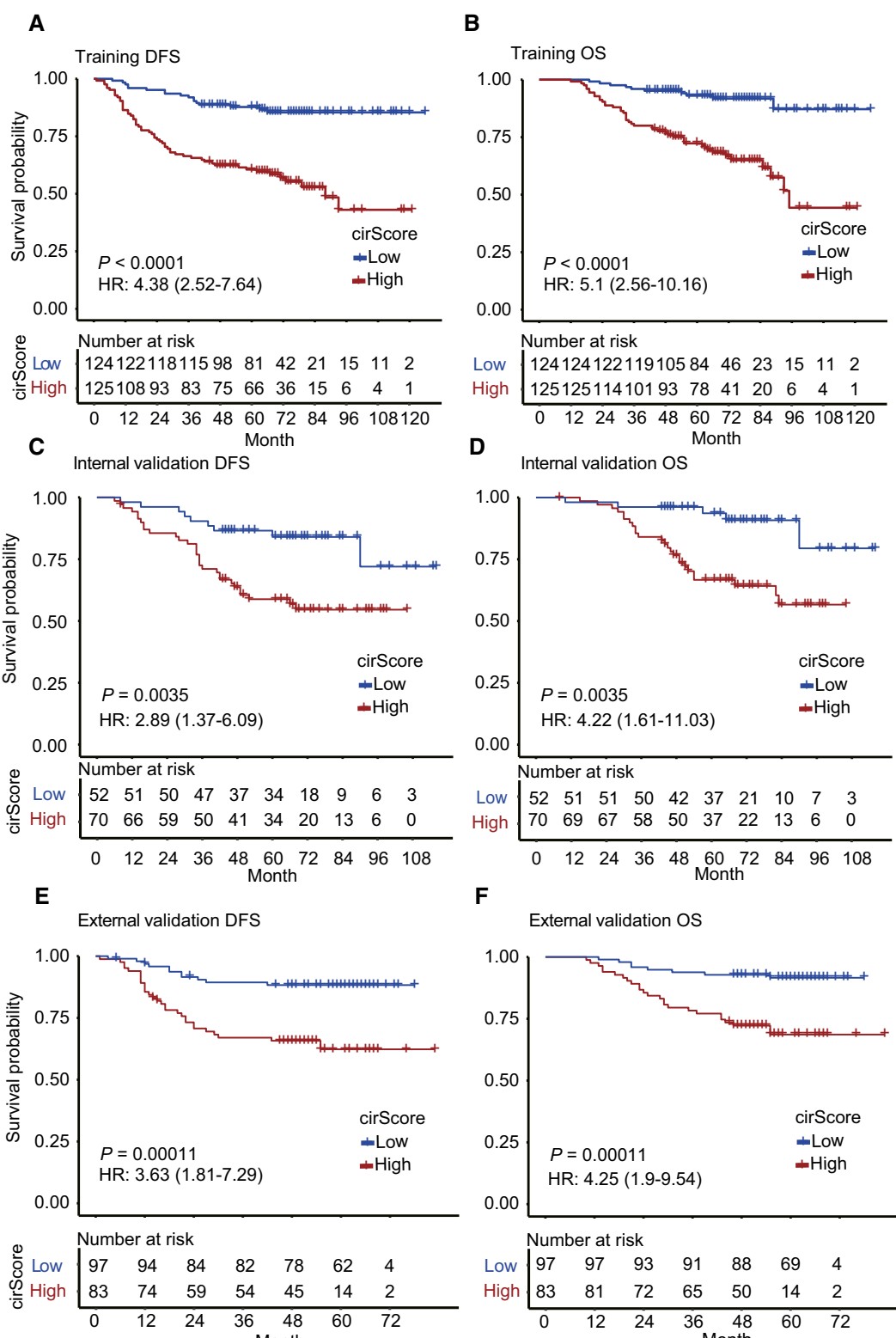

**Figure 3. Kaplan–Meier curves of DFS and OS based on the cirScore in patients with stage II/III colon cancer.**

A–F  Kaplan–Meier curves of DFS (A) and OS (B) in 249 patients in the training set. Kaplan–Meier curves of DFS (C) and OS (D) in 122 patients in the internal validation dataset. Kaplan–Meier curves of DFS (E) and OS (F) in 180 patients in the external independent validation dataset. Hazard ratios (HRs) were calculated with a univariate Cox regression analysis, and P-values were calculated with the log-rank test.

Source data are available online for this figure.

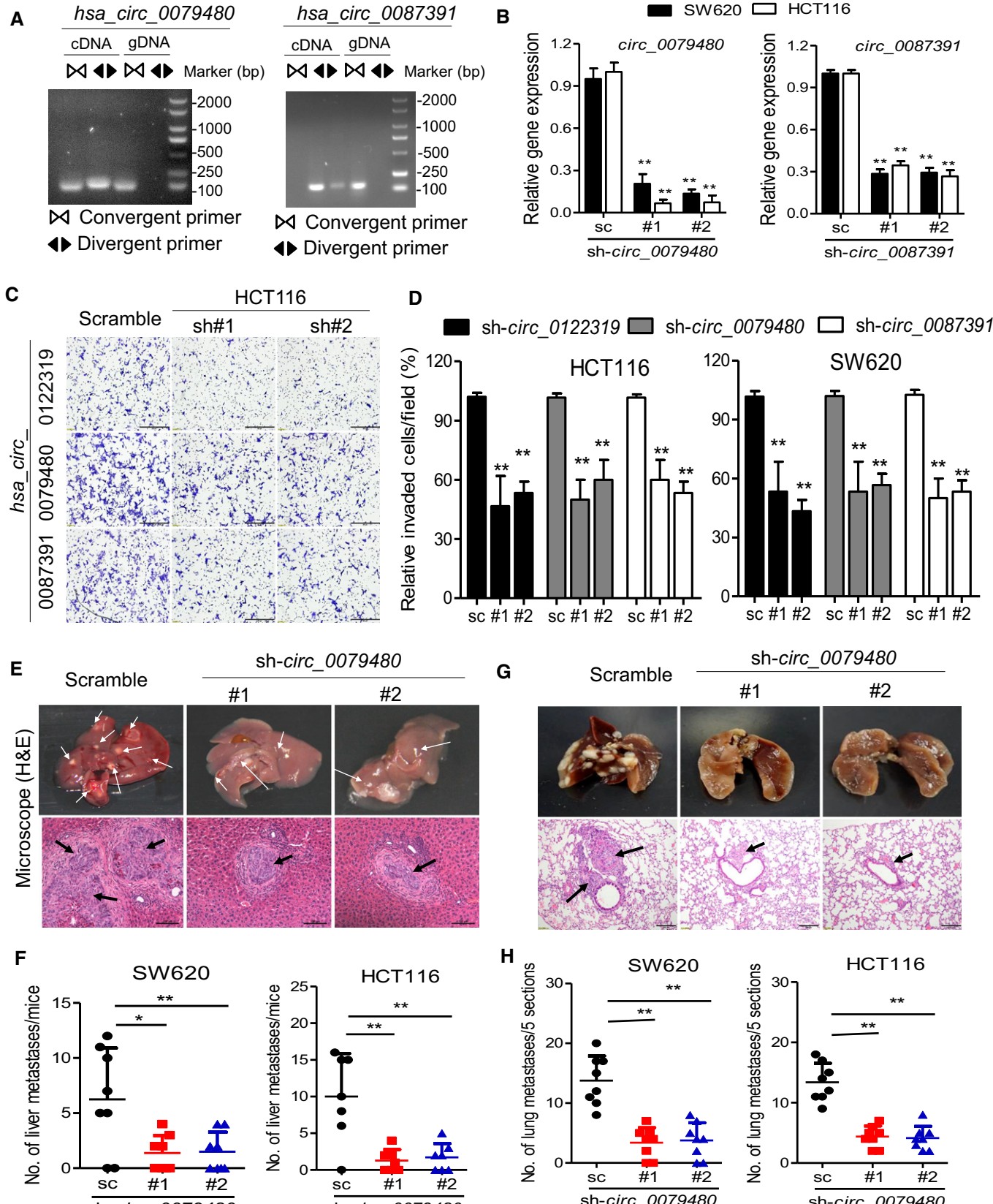

**Figure 4.**

◀

**Figure 4.  Loss-of-function assay of candidate circRNAs regulating cell invasion.**

A   RT–PCR products with divergent and convergent primers showing circularization of has_circ_0079480 and has_circ_0087391. cDNA, complementary DNA; gDNA, genomic DNA.

B   qRT–PCR evaluated the knockdown efficiency of has_circ_0079480 and has_circ_0087319 in SW620 and HCT116 cells transfected with two unique shRNAs (#1, #2). **$P < 0.01$, Student's $t$-test, mean $\pm$ SD ($n = 3$).

C   Representative images of the migration phenotype in HCT116 cells with knockdown of candidate circRNAs, scale bar: 100 μm.

D   The relative fold change of the transwell migration for indicated knockdown cells over those of control cells. **$P < 0.01$, Student's $t$-test, mean $\pm$ SD ($n = 3$).

E, F   Representative hematoxylin and eosin (H&E) staining and statistical results of the micro-metastatic nodules in the liver from mice injected with the indicated cells into the spleen for 45 days, white and black arrows indicate the liver metastatic foci, scale bar: 100 μm. $N = 8$ per group. *$P < 0.05$; **$P < 0.01$, Student's $t$-test, mean $\pm$ SD.

G, H   Representative H&E staining and statistical results of metastatic lung nodules from mice injected with the indicated cells via the tail vein for 60 days. Five sections evaluated per lung, black arrows indicate the lung metastatic foci, scale bar: 100 μm. $N = 8$ per group. **$P < 0.01$, Student's $t$-test, mean $\pm$ SD.

Data information: Exact $P$-values are specified in Appendix Table S5.

nomogram-predicted DFS among both validation sets (Fig EV5A and B). Moreover, a nomogram for predicting the 3-year or 5-year OS with the same variables was built (Fig EV5C and Appendix Table S3). The C-indices were 0.831 (95% CI, 0.780–0.881), 0.835 (95% CI, 0.773–0.897), and 0.769 (95% CI, 0.685–0.853), respectively, in the three datasets. Calibration plots suggested good consistency between the actual and nomogram-predicted OS probabilities in all three datasets (Fig EV5D). Time-dependent ROC analyses also indicated the superior prognostic accuracy of the nomograms for prediction of DFS and OS compared to the existing risk factors in all three datasets (Fig 5C–E). In the training set, the time-dependent AUC for the existing risk factors ranged from 0.52 (95% CI, 0.46–0.59) to 0.69 (0.62–0.76) for DFS and from 0.55 (95% CI, 0.47–0.62) to 0.73 (0.66–0.80) for OS, whereas the AUC for the proposed nomograms reached 0.85 (95% CI, 0.79–0.90) for DFS and 0.85 (95% CI, 0.79–0.91) for OS (Fig 5C). Time-dependent ROC analyses for the internal and external validation sets yielded consistent findings (Fig 5D and E). In summary, these results strongly suggest the clinical utility of the proposed nomograms for prediction of DFS and OS.

## Discussion

In this study, we developed and validated a novel prognostic tool based on four circRNAs to improve the prognostic stratification for patients with radically resected stage II/III colon cancer. Our results showed that this tool can effectively classify patients with stage II/III colon cancer into groups with low and high risks of disease recurrence. Furthermore, this proposed cirScore provided additional prognostic value to existing clinicopathological prognosticators for stage II/III colon cancer. Of particular importance, this is the first study that demonstrates the clinical utility of the circRNA signature as a postoperative prognostic tool in patients with stage II/III colon cancer.

For patients with R0 resected stage III or high-risk stage II colon cancer, adjuvant chemotherapy is considered a standard of care. However, previous evidence suggests that adjuvant chemotherapy, with or without oxaliplatin, conveyed limited benefits to patients with high-risk stage II disease (O'Connor et al, 2011). In contrast, adjuvant chemotherapy has shown a robust efficacy in patients with stage III disease and 6-month oxaliplatin-based chemotherapeutic regimens have become standard adjuvant treatment for these patients since 2004 (Andre et al, 2004). Given the cumulative neurotoxicity associated with oxaliplatin exposure, the International Duration Evaluation of Adjuvant Therapy (IDEA) collaboration conducted a prospective pooled analysis and showed that 3 months of adjuvant therapy

appeared to be sufficient in a lower-risk group (defined as patients with T1, T2, or T3/N1 disease), especially when the capecitabine and oxaliplatin combination was chosen. In a higher-risk group (patients with T4, N2, or both), 6 months of adjuvant therapy may be needed, particularly when the fluorouracil and oxaliplatin combination was the chosen regimen (Grothey et al, 2018). Notably, our study showed that patient survival was heterogeneous even within the high- or low-risk stage II/III groups; that is, patients with high-risk stage II and low- and high-risk stage III disease could be further stratified by the cirScore into subsets with distinct outcomes, suggesting a room for tailoring treatment strategies and avoiding overtreatment or undertreatment in selected patients. Moreover, we proposed prognostic nomograms that allow for individualized estimation of the 3- and 5-year DFS and OS probabilities among patients with radically resected stage II/III colon cancer. Taken together, the cirScore and the associated nomograms may serve as a clinically useful tool to improve surveillance and guide decision making regarding the administration of adjuvant chemotherapy and treatment duration.

Recently, the circRNA study has captured the interest of many in the scientific and medical communities (Vicens & Westhof, 2014; Ebbesen et al, 2017). Owing to their unique properties, such as being evolutionarily conserved, having tissue-specific expression, more stable than linear miRNA, circRNAs may serve as potential diagnostic or predictive biomarkers for colorectal cancer patients (Ebbesen et al, 2017; Taborda et al, 2017). Some circRNAs have been shown to be associated with prognosis and regulate cell biological function in colorectal cancer, including circHIPK3, circCCDC66, and CiRS-7 (Hsiao et al, 2017; Weng et al, 2017; Jiang et al, 2018; Zeng et al, 2018). A recent study also demonstrated the existence of abundant exo-circRNAs in the serum of colorectal cancer patients (Li et al, 2015). However, these studies have been limited by the small number of circRNAs screened, the small sample sizes, and the lack of independent validation. Our study included 667 patients and is therefore, to our knowledge, the largest circRNA-based biomarker discovery project to be done in stage II/III colon cancer. The use of the LASSO-based marker selection strategy and the Cox regression model allowed us to integrate multiple circRNAs into one tool, which has a significantly greater prognostic accuracy than that of single circRNAs alone. This method has been successfully applied to establish prognostic prediction models using other biomarkers, such as miRNA (Zhang et al, 2013) and circulating tumor DNA methylation (Xu et al, 2017). For the first time, we built a four-circRNA-based signature using the LASSO-bagging algorithm and the Cox regression model that can predict recurrence of stage II/III colon cancer patients. Among these four circRNAs, hsa_circ_0008039 has been

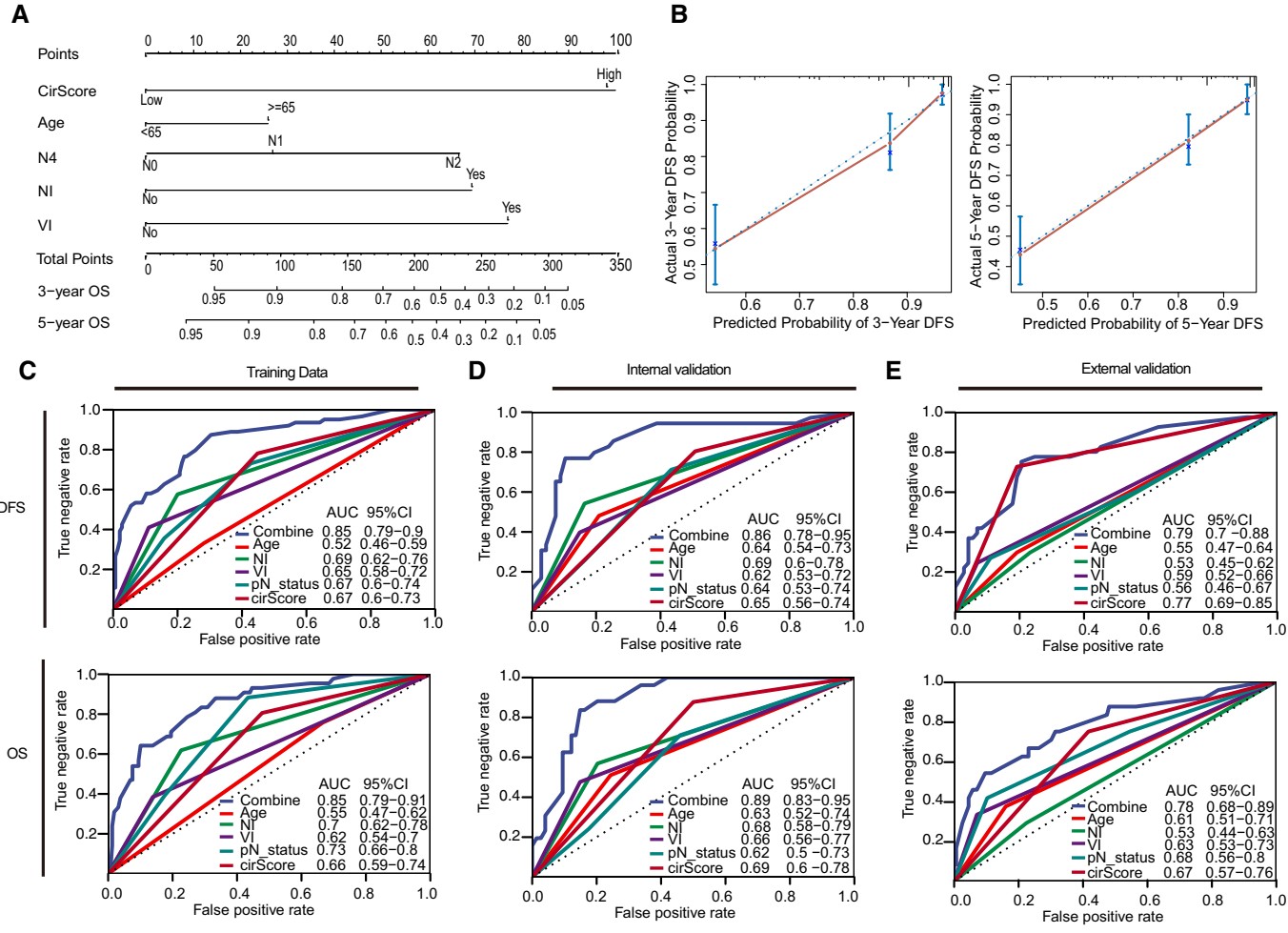

**Figure 5. Building nomogram and time-dependent ROC analysis.**

A   Nomogram to predict the DFS of patients with stage II/III colon cancer.

B   Calibration curves of the nomogram in prediction of the 3-year and 5-year DFS in training cohort (*n* = 249). The 45-degree blue dotted line represents the reference line of an ideal nomogram. The error bars represent the 95% confidence intervals of the actual survival in the upper, middle, and lower textiles. NI, perineural invasion. VI, lymphatic or vascular invasion.

C–E   Plots shown in the figures are DFS and OS prediction in the training set (C), internal validation set (D), and external validation dataset (E). AUC = area under the curve.

reported to promote breast cancer progression by regulating miR-432-5p/E2F3 axis (Liu *et al*, 2018). However, the other novel circRNAs have not been investigated in cancer.

Limitations of the present study should be acknowledged. The retrospective nature of this study made it susceptible to inherent biases. Additionally, in view of potential selection bias and limited sample size, we were not able to determine how the proposed cirScore and nomograms may modify treatment strategies for stage II/III colon cancer, and further prospective trials addressing this issue are needed. Moreover, this study was East Asia-centric and patient cohorts from other geographical regions are required to validate our findings. Despite these limitations, this study represents currently the best evidence regarding the potential clinical utility of circRNA-based signatures for prognostic stratification in patients with middle stage colon cancer, and directly quantifies circRNA expression from fresh colon cancer tissues based on qPCR assay, making it easy to implement in clinical practice.

In conclusion, the cirScore can effectively classify patients with radically resected stage II/III colon cancer into groups with different risks of recurrence, thereby raising the possibility that circRNAs may be supplementary to the traditional clinicopathological risk factors as a prognostic scheme. Additionally, the proposed nomograms incorporating the cirScore and existing clinical prognosticators might facilitate personalized postoperative surveillance and management of patients with stage II/III colon cancer.

# Materials and Methods

### Patient enrollment

We collected frozen tissue samples from patients who met all the following criteria: (i) histologically confirmed as stage II/III colon cancer between January 1, 2010, and December 31, 2013, according

to the 7th edition American Joint Committee on Cancer staging scheme; (ii) underwent histologically confirmed R0 resection; and (iii) availability of complete follow-up data. Patients were excluded if they had previous treatment with any anticancer therapy, had any tumor type other than adenocarcinoma or mucinous carcinoma, or insufficient RNA (< 5 ng/μl) available. Two pathologists (QN Wu, XJ Fan) reassessed all of the samples, all of which were found to contain more than 70% tumor cells. All the tissue samples were collected from patients with informed consent. Studies were conducted in alignment with the ethical principles for medical research involving human subjects set out in the World Medical Association Declaration of Helsinki and Department of Health and Human Services Belmont Report and were approved by the ethics committees of both participating institutions.

### Bioinformatic analysis

We retrospectively collected 20 paired of frozen tumor tissues and adjacent normal tissues of the primary site from patients with stage II/III colon cancer from the discovery set, including 10 recurrence and 10 non-recurrence patients within 5 years after surgery. To identify potentially deregulated circRNAs that correlated with the outcomes of the colon cancer patients, we conducted an RNA-sequencing study and profiled circRNAs by a series of bioinformatic analysis as described below. We employed limma (Ritchie *et al*, 2015) to identify differentially expressed circRNAs between the tumor and normal groups, or between the recurrence and non-recurrence groups, with a threshold of 1 for the log fold change and a *P*-value < 0.05.

For marker selection, we only considered overexpressed circRNAs for detection convenience, and 100 circRNAs were selected for further validation according to the following retaining criteria: (i) upregulated circRNAs; (ii) circRNAs located in the junction site of exons; and (iii) fold change > 5.0, *P*-value < 0.05, and the raw intensity of each sample > 200. We applied the qRT–PCR assay to validate the selected circRNAs in a larger cohort for further selection. circRNAs that were validated as having the same expression trend and that had a *P*-value < 0.05 were considered as consistent markers. We next tested those markers on the samples from the training and validation cohorts by qRT–PCR assay.

### Identification and quantification of circle RNAs from RNA-seq dataset

Raw RNA-sequencing reads of each sample were aligned to the hg38 human genome using TopHat2 software (Kim *et al*, 2013). Three bioinformatics circle RNA analytic methods, circRNA_finder (Westholm *et al*, 2014), CIRI (Gao *et al*, 2015), and UROBORUS (Song *et al*, 2016), were used for circRNA identification with default parameters. Next, we filtered out the circRNAs with less than two samples expressed. To annotate circle RNAs, we converted the hg38 coordinates of each circRNA into hg19 by using liftOver program from UCSC (Kent *et al*, 2002). The nearest protein-coding gene for a circRNA was determined according to the distance from the corresponding circRNA along the genome sequence. All known circRNAs were named with circBase ID referring to circBase annotation. Novel circRNAs were named according to their rank number summarized in final table.

To classify, all circRNAs were divided into seven types after intersection with known transcript (Memczak *et al*, 2013): exonic circRNA, intronic circRNA, 3′UTR circRNA, 5′UTR circRNA, antisense circRNA, intergenic circRNA, and ncRNA circRNA. Expression levels of circRNAs were quantified by the number of junction-spanning reads obtained from the UROBORUS tool. The Transcripts Per Million (TPM) of reads of circRNAs were calculated to obtain an estimate of relative expression. The circRNAs with a *P*-value ≤ 0.05 and an absolute value of $\log_2$ fold change ≥ 1 were treated as differentially expressed.

### qRT–PCR assay

Total RNA was isolated with TRIzol reagent (#15596-08, Life Technologies, Carlsbad, USA) and then reverse-transcribed with random hexamers using a PrimeScript RT Reagent Kit (TaKaRa Bio, Inc., Shiga, Japan) according to the manufacturer's protocol. The resulting complementary DNA was analyzed by qRT–PCR performed with SYBR reagent using the IQ5 PCR system (Bio-Rad, Hercules, CA). β-Actin was used as the internal control gene, and data were analyzed using the $2^{-\Delta\Delta ct}$ method. Specific divergent primers, convergent primers, and primers for detecting the corresponding host genes were designed by Geneseed Biotech. (Guangzhou, China), synthesized by Sigma-Aldrich (Louis, MO, USA). These primer sequences are described in Appendix Supplementary Methods. The circRNA ID, gene symbol, and back-splice junction (BSJ) coordinate for 22 circRNAs are described in Appendix Table S4.

### LASSO-bagging procedure

The qRT–PCR values of each circRNA were scaled to a reasonable range with the following equation: $Exp = Log_2$ (*Relativevalue* × $10^3$). Outliers were replaced by NAs if they had extremely high (< 1.5× interquartile range [IQR]) or low values (< 1.5 × IQR). The matrix was then imputed with K-nearest neighbor imputation (Beretta & Santaniello, 2016).

To further narrow down the candidate list, we first filtered out the circRNAs with Wald $P ≥ 0.05$ by univariate Cox regression analysis with disease-free survival, and 8 circRNAs were remained in the training set. DFS and the expression matrices of validated circRNAs were then subjected to the LASSO-bagging procedure. LASSO is a popular method for regression with high dimensional predictors (Tibshirani, 1997), and broadly applied to the Cox proportional hazard regression model for survival analysis (Zhang *et al*, 2013). Here, we applied a multisplit strategy with LASSO to reduce the overfitting from the training dataset as described previously (Xu *et al*, 2017). The algorithm contains the following steps: (i) bootstrapping the data point 500 times and generated 500 training matrices; (ii) for each matrix with PFS, Lasso Cox regression analysis was performed using 10-fold cross-validation. Tuning parameter λ was chosen by 1-SE (standard error), and we finally got a list of variables that had non-zero beta coefficient in Lasso fit output; (iii) collapse all variable list obtained in each matrix and the resample model inclusion proportion (RMIP) for each circRNA was calculated (explained by an observed frequency in 500 resamples); and (iv) using RMIP as weight of each variable, we observed a sharp RMIP decrease after the fourth marker when ranked all markers in a decrease order. We finally selected the top four markers to build the regression model.

## Characterization of the selected circRNAs

Total RNA (4 μg) was isolated with TRIzol reagent (Cat. # 15596-08, Life Technologies, Carlsbad, USA) and incubated with 3 U/μg of RNase R (Epicentre Technologies, Madison, WI, USA) for 15 min at 37°C or mock-treated. The RNA was immediately transferred to ice, spiked with 10% mouse RNA, and extracted with TRIzol reagent. The RNA concentration of the control group was determined, and the same volume (6 μl) of control RNA and the RNase R-treated RNA was used for reverse transcription. The circRNA expression levels and corresponding host genes' mRNAs were analyzed by qRT–PCR quantification, and the mouse GAPDH mRNA was used for normalization as an exogenous control as reported (Zhang *et al*, 2016; Pamudurti *et al*, 2017). The detected primer sequences for the corresponding host genes and mouse gapdh were described as follows: ISPD (forward: 5′-cccaccccgaagcaattct-3′, reverse: 5′-tccaac atactctctccaggg-3′); AGTPBP1 (forward: 5′-tctaggatcgtaggactcctgg-3′, reverse: 5′-acatatcgggcagtgtctgat-3′); PLOD2 (forward: 5′-catggacaca ggataatggctg-3′, reverse: 5′- aggggttggttgctcaataaaaa-3′); PRKAR1B (forward: 5′-caggtcctcaaagactgtatcgt-3′, reverse: 5′-atgggagtccgactg tgagt-3′); mouse gapdh (forward: 5′-aggtcggtgtgaacggatttg-3′, reverse: 5′-ggggtcgttgatggcaaca-3′).

Besides, specific convergent primers were designed by Geneseed Biotech. (Guangzhou, China), synthesized by Sigma-Aldrich (Louis, MO, USA), and the sequences were described as follows: hsa_ circ_0079480 (forward: 5′-catctgaggctctgggtcat-3′, reverse: 5′-tgggtttct tgaaaatcagagg-3′); hsa_circ_0087391 (forward: 5′-ctcccccacatgaggactt a-3′, reverse: 5′-ctgcaaattctgcttgacca-3′); hsa_circ_0122319 (forward: 5′-ggattccatcgatttatgcag-3′, reverse: 5′-ctggcccctccaatacta-3′).

## Cell culture and migration assay

The human CRC cell lines and immortalized/non-tumorigenic cells were purchased from the ATTC (Manassas, VA, USA) and cultured under conditions specified by the supplier. All cells were negatively tested for mycoplasma contamination before use, and authenticated based on STR fingerprinting before use at Medicine Lab of Forensic Medicine Department of Sun Yat-sen University. The lentiviruses containing shRNA targeting circRNAs were purchased from Gene-Pharma (Shanghai, China), and the lentiviral transduction was performed as previously described (Ju *et al*, 2017). The shRNA sequences targeting against the circRNAs were as follows: hsa_ circ_0122319 (#1: atgtttactgaatgataaatt; #2: ctgaatgataaaattattagtc); hsa_circ_0079480 (#1: gttgttgtttcaagagaattt; #2: gtttcaagagaatttccc aag); hsa_circ_0087391 (#1: cagtcttataaaattatctgc; #2: cttataaaattatc tgcaatt). Then, cell migration assay was conducted to assess the *in vitro* function of the selected circRNAs as previously described (Ju *et al*, 2016). To rule out off-target effects, RNA-seq and bioinformatic analyses were performed by the Novogene Corporation (Beijing, China).

## Immunoblotting analysis

Immunoblotting analysis was conducted with standard procedures as previously described (Ju *et al*, 2017). Briefly, cells were lysed in RIPA buffer and normalized using a BCA Protein Assay Kit (Thermo Scientific, Waltham, MA, USA). Proteins were separated by SDS–PAGE and blotted onto a PVDF membrane (Millipore, Billerica, MA,

USA). Membranes were probed with the specific primary antibodies and then with peroxidase-conjugated secondary antibodies. β-Actin antibody was used as a loading control. The bands were visualized by enhanced chemiluminescence using Hyperfilm ECL. The following antibodies were used for immunoblotting analysis: PLOD2 antibody (1:800, #ab72939) (Abcam, Cambridge, MA, USA) and β-actin (1:1,000, #3700) (Cell Signaling, Danvers, MA, USA).

## *In vivo* metastasis study

Two xenograft models were used to evaluate the *in vivo* metastasis effects of circ_0079480 that exhibited *in vitro* function as previously described (Ju *et al*, 2018). Female BALB/c nude mice (3/4 weeks old) were obtained from the Animal Center of Guangdong Province (Guangzhou, China) and housed under specific pathogen-free (SPF) conditions. For liver metastasis, the cells ($2 \times 10^6$) in 50 μl PBS were injected into the distal tip of the spleen using a Hamilton syringe (8 mice/group). Six weeks later, the mice were sacrificed and the spleen and liver were removed and embedded in paraffin. The numbers of metastatic nodules in the livers were counted. For tumor lung metastasis, the circ_0079480 knockdown and control cells ($2 \times 10^6$) in 100 μl PBS were injected into the tail vein of nude mice (8 mice/group). Six weeks postinjection, the mice were killed and the lung was removed and paraffin-embedded. Consecutive sections were made and stained with hematoxylin and eosin (H&E). The micrometastases in the lungs were examined and counted under a dissecting microscope.

All animal experiments were performed in accordance with a protocol approved by our institutional Animal Care and Use Committee. The randomization of animal allocation was done by random numbers generated by the computer. Following experimentation, no animals were excluded from analysis, and no blinding procedure was undertaken. The reporting of mouse studies in this manuscript conforms with the Animal Research: Reporting of *In Vivo* Experiments (ARRIVE) guidelines (Kilkenny *et al*, 2010).

## Statistical analysis

For survival analyses, we used the Kaplan–Meier method to analyze the correlation between variables and the survival, and the log-rank test to compare between-group survival. We used the Cox regression model to do the multivariable survival analysis, and Cox regression coefficients to generate nomograms. Concordance indices (C-indices) were used to measure the discriminative abilities of the nomograms (Harrell *et al*, 1996). Calibration was performed by reviewing the plots of nomogram-predicted survival probabilities with the Kaplan–Meier-estimated probabilities (Iasonos *et al*, 2008). All statistical tests were two-sided, and $P < 0.05$ was deemed significant. All analysis scripts were programmed using R software (v3.3.3), with the "glmnet" package (R Foundation for Statistical Computing, Vienna, Austria) for LASSO, the "rms" package for development of nomogram, and the "survival ROC" package to do the time-dependent ROC curve analysis.

For functional assay, all experiments that were repeated three times are presented as mean ± standard deviation (SD), evaluated using Student's *t*-test (unpaired, two-tailed). Sample size was chosen based on the need for statistical power. Differences reached

### The paper explained

**Problem**

Current staging methods seem to have only a limited role in predicting the risk of disease recurrence and benefit of adjuvant chemotherapy for patients with stage II/III colon cancer. Circular RNA is a novel type of non-coding RNA with a potential use as biomarkers; however, whether circRNA-based signatures could serve as novel prognostic biomarkers for stage II/III colon cancer is unknown.

**Results**

Dysregulated circRNAs showed strong classification properties in distinguishing the recurrent colon cancer patients from non-recurrent colon cancer patients. A novel prognostic tool (cirScore) based on four circRNAs (i.e., hsa_circ_0122319, hsa_circ_0087391, hsa_circ_0079480, and hsa_circ_0008039) is developed and validated to improve the prognostic stratification for patients with radically resected stage II/III colon cancer. The proposed cirScore can effectively classify patients with stage II/III colon cancer into groups with low and high risks of disease recurrence. Loss-of-function assays indicated that the representative circRNAs play functional roles in the sophisticated regulation of colon cancer progression.

**Impact**

Our current study addresses an important gap, which is the refinement of our prognostic tools for stage II/III colon cancer, by using a novel approach that takes into consideration the circular RNA. The proposed cirScore might be used in the future to guide better and more personalized treatment decisions for patients with stage II/III colon cancer.

statistical significance with $P < 0.01$ (**) and $P < 0.05$ (*), analyzed by GraphPad Prism 5 (La Jolla, CA, USA).

## Data availability

RNA-seq data of cell lines are available on Sequence Read Archive (SRA) PRJNA551560 (https://www.ncbi.nlm.nih.gov/bioproject/?term = PRJNA551560). RNA-seq data for CRC samples are available on Sequence Read Archive (SRA) PRJCA001113 (http://bigd.big.ac.cn/gsa-human/). Modeling computer scripts are available on GitHub (https://github.com/likelet/CircRNA_colon_recurrent_prediction, https://github.com/likelet/RNAseqPipe).

**Expanded View** for this article is available online.

## Acknowledgments

This research was supported by National Natural Science Foundation of Guangdong Province (2018B030306049, 2014A030312015); Pearl River S&T Nova Program of Guangzhou (201806010002); Health Medical Collaborative Innovation Program of Guangzhou (201803040019, 201704020228); Science and Technology Planning Project of Guangdong Province (A2019398, 2015B020232008); the National Natural Science Foundation of China (81930065).

## Author contributors

Conception and design: R-HX, H-QJ; Development of methodology: H-QJ, QZ, Z-XZ; Acquisition of data (provided animals, acquired and managed patients, provided facilities, etc.): FW, PL, ZW, Q-NW, X-JF, H-YM, LC, TL, CR, X-BW, GC, Y-HL, W-HJ; Analysis and interpretation of data (e.g., statistical analysis, biostatistics, computational analysis): H-QJ, QZ, FW, PL, Z-XZ, ZW, LC; Writing, review, and/or revision of the manuscript: R-HX, H-QJ, QZ, ZW, TL; Administrative, technical, or material support (i.e., reporting or organizing data, constructing databases): R-HX, H-QJ; Study supervision: R-HX.

## Conflict of interest

The authors declare that they have no conflict of interest.

## For more information

The URLs for data presented in this article are as follows:

(i)   circBase, http://www.circbase.org/
(ii)  circRNA_finder: https://github.com/orzechoj/circRNA_finder
(iii) CIRI: https://sourceforge.net/projects/ciri
(iv)  UROBORUS: http://uroborus.openbioinformatics.org

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
