## [Review Process File · EMBO Molecular Medicine]

A circRNA signature predicts postoperative recurrence in stage II/III colon cancer

Huai-Qiang Ju, Qi Zhao, Feng Wang, Ping Lan, Zixian Wang, Zhi-Xiang Zuo, Qi-Nian Wu, Xin-Juan Fan, Hai-Yu Mo, Li Chen, Ting Li, Chao Ren, Xiang-Bo Wan, Gong Chen, Yu-Hong Li, Wei-Hua Jia, Rui-Hua Xu

Review timeline:	Submission date:	5 December 2018
	Editorial Decision:	11 January 2019
	Revision received:	20 March 2019
	Editorial Decision:	14 May 2019
	Revision received:	28 June 2019
	Editorial Decision:	7 August 2019
	Revision received:	15 August 2019
	Accepted:	16 August 2019

Editor: Céline Carret

Transaction Report:

1st Editorial Decision

11 January 2019

Thank you for the submission of your manuscript to EMBO Molecular Medicine. We have now heard back from the two referees whom we asked to evaluate your manuscript.

As you will see from the comments below, while ref. 2 is supportive of publication and only requests to use more animals, move a SI figure as main and test the diagnosis signature using different circRNA combination, ref. 1 is more critical. This referee regrets the lack of conclusiveness and controls should be provided along with better description of techniques and programs used. The authors should deposit their data in public repositories (this is mandatory for publication in any EMBO Press journals), and reorganise the structure of the paper.

We would therefore welcome the submission of a revised version within three months for further consideration and would like to encourage you to address all the criticisms raised as suggested to improve conclusiveness and clarity. Please note that EMBO Molecular Medicine strongly supports a single round of revision and that, as acceptance or rejection of the manuscript will depend on another round of review, your responses should be as complete as possible.

I look forward to receiving your revised manuscript.

***** Reviewer's comments *****

Referee #1 (Remarks for Author):

manuscript entitled: "A circRNA signature predicts postoperative recurrence in stage II/III colon

cancer" Xu et al., utilize circRNA expression to biomark colon cancer. Briefly the authors profile circRNAs from 667 patients with R0 resected stage II/III colon cancer. They then utilize a computational approach to identify a signature of circRNAs that could predict postoperative recurrence. They then validated the classifier in internal and external cohort. Last but not least, the performed loss of function assays that suggest a function of the selected circRNAs in colon cancer. This study addresses two very important and timely topics: putative roles of circRNAs in cancer and the possibility of using these molecules to biomark cancer. Unhappily the manuscript is below threshold for publication in EMBO Molecular Medicine and probably in most journals. However, if the authors can comprehensively modify the manuscript, in particular perform the required controls, it will be suitable for publication

Major issues:

1. None of the circRNAs under study have been verified for circularity. This should be done by RNaseR pretreatment followed by northern blot or qPCR with internal spike in. Moreover, the authors must state which circRNA annotation are they using. As I could check in circBase for the 4 circRNAs that form the cirScore, hsa_circ_0079480; hsa_circ_0008039; hsa_circ_0087391 were previously studied and annotated. However, circRNA13452 was impossible to find. The authors must use circBase annotation and provide all the information regarding this circRNAs, and the validated 22, (gene symbol, back-splice junction coordinates and previous reports on them) in a supplementary table.
2. The shRNA experiments are not quality for publication. No verification of the knockdown or the specificity was performed (not even against the host mRNA). The authors should indeed use more shRNAs (at least 3), show that all give the phenotype, show that all downregulate the targeted shRNA, show no effect on the linear mRNA (even better if possible at the protein level, by western blot) and perform a computational analysis to rule out off targets.
3. There are no GEO accession numbers or other public repository accession to the RNA sequencing data. In addition, the differential expression results must be complete in a supplementary table.
4. The authors do not state which circRNA search tool they use (find_circ1 or 2, circ_finder, CIRI, etc) or even which genome annotation they used.
5. The authors must state from which publication they are taking the least absolute shrinkage and selection operator (LASSO), LASSO-bagging algorithm, and how are they implementing it. They refer in Page 5: "model-based LASSO-bagging screen strategy" with no citation. They refer in Page 12: "This method has been successfully applied to establish prognostic prediction models using other biomarkers, such as mRNA (Yamanaka et al, 2016), miRNA (Zhang et al, 2013) and circulating tumor DNA methylation (Xu et al, 2017)". However, in Yamanaka et al no LASSO was explicitly used. On the contrary, in Zhang et al they explicitly state "We used the R software version 3.0.1 and the "glmnet" package (R Foundation for Statistical Computing, Vienna, Austria) to do the LASSO Cox regression model analysis.". What is more, in supplementary methods they give detail explanation on the LASSO. Huai-Qiang Ju et al, should do something similar.
6. The authors must discuss in detail the results of figure 5. They only mention "ROC analyses also indicated the superior prognostic accuracy of the nomograms for prediction of DFS and OS compared to the existing risk factors in all three datasets (Figure 5A-5F)". Also they should consider combine them with figure 4 or create a whole new results section talking about figure 5.
7. The authors should consider reorganizing the text and putting "Loss-of-function assay of selected circRNAs regulating cell metastasis" section before "Stratified analysis with known risk scheme" section.

Minor issues:

In the abstract:

"Using RNA-seq analysis, we profiled differential circRNA expression between patients with and without recurrence" They should say which tissue they sequenced and when.

"The four-circRNA-based cirScore was generated ". The phrase is not comprehensible by its own. The authors should say here something like: "With this information we generated a four-circ RNA..."

"shorter DFS and OS" the authors should define the acronyms they use before using them for the first time.

Page 3:

"and stage III colon cancer ." They have to define what are the marks for stage III as they did for stage II.

Page 4:

"The corresponding 5-year DFS (...) and 5-year OS rates" The authors have to define the terms they use (DFS and OS in this case).

Page 6:

"Risk score = (0.46expression level of circRNA13452)+(-0.386expression level of hsa_circ_0087391)+(0.293expression level of hsa_circ_0079480)+(0.439expression level of hsa_circ_0008039)" should be written as formula, not text.

Page 9:

The "RNAi assay" expression is misleading because the authors then talk about shRNA and this is a slightly different approach.

Page 24:

"Representative H&E staining and statistical results of metastatic lung nodules from mice injected with the indicated cells via the tail vein for 60 days" The authors must state what is H&E.

Figure 6:

The titles in the left are not centered.

Figure 1:

The authors should consider a different approach to explain the experimental procedure. Different colors and more space between the text are needed.

Referee #2 (Remarks for Author):

In this manuscript, authors characterized a circRNA-based signature in prognostic evaluation of colon cancer. They found that cirScore of four circRNAs could be used to define colon cancer patients as high- and low-risk groups. Patients in high-risk group (with high cirScore) had a poorer DFS and OS than patients in low-risk group. This is a very interesting study.

Comments and suggestions:

- 1) In Figure 2D, 4 circRNAs are strongly predictive of DFS. Have authors tried to use a single circRNA (e.g. circRNA13452) or combination of two or three circRNAs to predict the prognosis of patients with colon cancer (e.g. circRNA13452 and hsa_circ_0087391)? What are the results?
- 2) In page 9 and page 10, the number of tested nude mice should be 8 per group.
- 3) As the result of knockdown of circRNAs (Figure S6B) is important, I suggest authors to move this data to Figure 6.

1st Revision - authors' response

20 March 2019

Point-by-point replies to the reviewers' comments

Referee #1 (Remarks for Author):

Manuscript entitled: "A circRNA signature predicts postoperative recurrence in stage II/III colon cancer" Xu et al., utilize circRNA expression to biomark colon cancer. Briefly the authors profile circRNAs from 667 patients with R0 resected stage II/III colon cancer. They then utilize a computational approach to identify a signature of circRNAs that could predict postoperative recurrence. They then validated the classifier in internal and external cohort. Last but not least, the performed loss of function assays that suggest a function of the selected circRNAs in colon cancer.

This study addresses two very important and timely topics: putative roles of circRNAs in cancer and the possibility of using these molecules to biomark cancer. Unhappily the manuscript is below threshold for publication in EMBO Molecular Medicine and probably in most journals. However, if

the authors can comprehensively modify the manuscript, in particular perform the required controls, it will be suitable for publication

Response: Thank you very much for your review of our manuscript. We also thank the reviewer for the positive comments regarding the scientific contribution of our manuscript and the following constructive suggestions.

Major issues: 1. None of the circRNAs under study have been verified for circularity. This should be done by RNaseR pretreatment followed by northern blot or qPCR with internal spike in. Moreover, the authors must state which circRNA annotation are they using. As I could check in circBase for the 4 circRNAs that form the cirScore, hsa_circ_0079480; hsa_circ_0008039; hsa_circ_0087391 were previously studied and annotated. However, circRNA13452 was impossible to find. The authors must use circBase annotation and provide all the information regarding this circRNAs, and the validated 22, (gene symbol, back-splice junction coordinates and previous reports on them) in a supplementary table.

Response: Thanks for pointing out this issue! As suggested, the circularity and stability of the four selected circRNAs have been verified by Sanger sequencing and RNase R treatment. After examined by RT-PCR with divergent primers, the sequenced PCR product was corresponding from the bioinformatics analysis with the exact back-splice junction (Fig EV2A). Compared with linear host genes, more resistance to digestion with RNase R exonuclease confirmed that these circRNAs harbors a circular RNA structure (Fig EV2B). These result has been added in the Fig EV2 and described in the revised manuscript (Page 8, Paragraph 1).

For the circRNA annotation, we firstly used the circBase for the known circRNAs, the novel identified and unannotated circRNAs in “circBase” are defined by rank number. The related information for these known and unannotated circRNAs has been provided in SourceDataForFig2 attached with Fig 2. For the circBase updates the gene list constantly, we just checked those previously unannotated circRNAs against the recent updated circBase. Surprisingly, we found that circRNA13452 (chr3:145838898-145842016), located on the chromosome chr3q24, has been included in the circBase annotated as hsa_circ_0122319. To maintain consistency, we have updated and modified circRNA13452 as hsa_circ_0122319 in our revised manuscript.

Besides, we also provided the more detailed information for validated 22 circRNAs, including gene symbol, divergent primer, back-splice junction coordinates and previous reports in Appendix Table S4.

2. The shRNA experiments are not quality for publication. No verification of the knockdown or the specificity was performed (not even against the host mRNA). The authors should indeed use more shRNAs (at least 3), show that all give the phenotype, show that all downregulate the targeted shRNA, show no effect on the linear mRNA (even better if possible at the protein level, by western blot) and perform a computational analysis to rule out off targets.

Response: We apologize for not providing convincing data. For these four circRNAs are generated by pre-mRNA 'back-splicing' of exons, we can only design two 21nt shRNAs targeting junctional sequences spanning 10–15 nt on either side. We also performed a computational analysis to rule out off targets. Take hsa_circ_0079480 for example as following:

(hsa_circ_0079480: The highlight letters “**TTTC**” are the circRNA back-splice junction site; Underlined letter are the sequences of shRNA#1, and the blue letters are the sequences of shRNA#2)

```
TTCAGCAAATCATCTTAGATCAATGCTACAATTTTGTGTTGTGAATGTTACAACCTCTG
ATTTTCAAGAAACCCAGAAGTTACTGAGCATGCTTGAAGAGAGTAGTCTTTGCATTTTA
TATCCTGTTGTTGTTTTTCTTTCGAGAATTTCCCAAGAGATTTGTGTAGTTATGGATACA
GAAGAAGATAACAAACATGTAGGTCATCTTCTTGAAGAAGTGCTGAAAAGTGAATTA
ATCATGTAAAAAGTCACATCTGAGGCTCTGGGTCATGCTGGCAGACATC
```

Both the knockdown efficiency and specificity were verified by qRT-PCR for the expression of circRNA and host mRNA (Figs 4B and EV3C). We also detected the protein expression encoded by three host genes (PLOD2, AGTPBP1, ISPD) in colon cancer cells with circRNA knockdown via immunoblotting analysis, however only PLOD2 protein be detected in colon cancer cells (Fig EV3D). The results demonstrated that knockdown of these circRNAs had no effects on the mRNA or protein expression of the host genes, suggesting that the regulatory effects directly results from targeting the circRNAs. These result has been described in the revised manuscript (Page 10, Paragraph 1). Hopefully, our above response gives you satisfaction!

3. There are no GEO accession numbers or other public repository accession to the RNA sequencing data. In addition, the differential expression results must be complete in a supplementary table.

Response: We thank the reviewer for the suggestion. We have submitted the RNA sequencing data to The Genome Sequence Archive for Human (GSA-Human) (HRA000037, Transcriptome profiles of patient with middle stage colon cancer). Also, the differential expression results have been provided in in SourceDataForFig2 attached with Fig 2.

4. The authors do not state which circRNA search tool they use (find_circ1 or 2, circ_finder, CIRI, etc) or even which genome annotation they used.

Response: We apologize for not providing a clear description. Actually, we applied three circRNA identification tools to search circRNAs, including circRNA_finder, CIRI and UROBORUS. In addition, the hg38 genome reference we employed in alignment step. Coordinates of circRNAs were converted into hg19 position with liftover program from UCSC, and sequentially used for circRNA annotation with circBase. We have added these detailed descriptions in Appendix Supplementary Methods.

5. The authors must state from which publication they are taking the least absolute shrinkage and selection operator (LASSO), LASSO-bagging algorithm, and how are they implementing it. They refer in Page 5: "model-based LASSO-bagging screen strategy" with no citation. They refer in Page 12: "This method has been successfully applied to establish prognostic prediction models using other biomarkers, such as mRNA (Yamanaka et al, 2016), miRNA (Zhang et al, 2013) and circulating tumor DNA methylation (Xu et al, 2017)". However, in Yamanaka et al no LASSO was explicitly used. On the contrary, in Zhang et al they explicitly state "We used the R software version 3.0.1 and the "glmnet" package (R Foundation for Statistical Computing, Vienna, Austria) to do the LASSO Cox regression model analysis". What is more, in supplementary methods they give detail explanation on the LASSO. Huai-Qiang Ju et al, should do something similar.

Response: We apologize for not providing a clear description. We have remove the wrong cited literature (Yamanaka et al, 2016), and added more detailed description and correct literatures in the "LASSO-bagging procedure" and "Statistical analysis" sections in revised manuscript (Methods, Page 18& Page 21).

6. The authors must discuss in detail the results of figure 5. They only mention "ROC analyses also indicated the superior prognostic accuracy of the nomograms for prediction of DFS and OS compared to the existing risk factors in all three datasets (Figure 5A-5F)". Also they should consider combine them with figure 4 or create a whole new results section talking about figure 5.

Response: We thank the reviewer for the suggestion. We have combined the ROC analyses results (previous Figure 5A-5F) with Figure 4 as revised Fig 5, and discuss this point more detailed in the revised manuscript (Page 13, Paragraph 1).

7. The authors should consider reorganizing the text and putting "Loss-of-function assay of selected circRNAs regulating cell metastasis" section before "Stratified analysis with known risk scheme" section.

Response: We thank the reviewer for the suggestion. We have reorganized the texts and the Figures as suggested in our revised manuscript, which read more smoothly.

Minor issues:

1) In the abstract: "Using RNA-seq analysis, we profiled differential circRNA expression between patients with and without recurrence" They should say which tissue they sequenced and when.

2) "The four-circRNA-based cirScore was generated". The phrase is not comprehensible by its own. The authors should say here something like: "With this information we generated a four-circ RNA..."

Response: Thanks for suggestions! We have modified these description in our revised Abstract as following, "Using RNA-seq analysis of 20 paired frozen tissues collected post-operation, we profiled differential circRNA expression between patients with and without recurrence, followed by quantitative validation. With clinical information, we generated a four-circRNA-based cirScore to classify patients into high-risk and low-risk groups in the training cohort."

3) "shorter DFS and OS" the authors should define the acronyms they use before using them for the first time.

Response: Thanks for pointing out this issue! We have defined the acronyms of “DFS and OS” as “disease-free survival (DFS) and overall survival (OS)” in our revised Abstract.

4) Page 4: "The corresponding 5-year DFS (...) and 5-year OS rates" The authors have to define the terms they use (DFS and OS in this case).

5) Page 3: "and stage III colon cancer." They have to define what are the marks for stage III as they did for stage II.

Response: Thanks for pointing out this issue! The acronyms and the terms of “DFS and OS” have been defined the first time they are used in the revised manuscript (Page 6, Paragraph 3). The term of “stage III colon cancer” has also been defined as N1/N2M0 disease irrespective of T stage in the revised manuscript (Page 5, Paragraph 1).

6) Page 6: "Risk score = (0.46expression level of circRNA13452)+(-0.386expression level of hsa_circ_0087391)+(0.293expression level of hsa_circ_0079480)+(0.439expression level of hsa_circ_0008039)" should be written as formula, not text.

7) Page 9: The "RNAi assay" expression is misleading because the authors then talk about shRNA and this is a slightly different approach.

Response: Thanks for pointing out this issue! The description “Risk score =...” has been changed as formula:

$$cirScore = 0.46 \times \text{Exp}_{hsa_circ_0122319} + 0.386 \times \text{Exp}_{hsa_circ_0083791} + 0.293 \times \text{Exp}_{hsa_circ_0079480} + 0.439 \times \text{Exp}_{hsa_circ_0008309}$$

in our revised manuscript (Page 8, Paragraph 2), and the “RNAi assay” has been modified as “lentivirus-mediated stable gene silencing” in our revised manuscript (Page 10, Paragraph 1).

8) Page 24: "Representative H&E staining and statistical results of metastatic lung nodules from mice injected with the indicated cells via the tail vein for 60 days" The authors must state what is H&E.

9) Figure 6: The titles in the left are not centered.

Response: Thanks for pointing out this issue! The abbreviation “H&E” has been stated as hematoxylin and eosin (H&E) in the revised manuscript (Page 20 & Page 28), and the left titles have been centered in the revised Fig 4.

10) Figure 1: The authors should consider a different approach to explain the experimental procedure. Different colors and more space between the text are needed.

Response: We thank the reviewer for the suggestion. We have restructured the flowchart and added some details to make it more readable as shown in revised Fig 1.

Referee #2 (Remarks for Author):

In this manuscript, authors characterized a circRNA-based signature in prognostic evaluation of colon cancer. They found that cirScore of four circRNAs could be used to define colon cancer patients as high- and low-risk groups. Patients in high-risk group (with high cirScore) had a poorer DFS and OS than patients in low-risk group. This is a very interesting study.

Response: Thank you very much for your review of our manuscript. We also thank the reviewer for the positive comments and the following constructive suggestions.

Comments and suggestions:

1) In Figure 2D, 4 circRNAs are strongly predictive of DFS. Have authors tried to use a single circRNA (e.g. circRNA13452) or combination of two or three circRNAs to predict the prognosis of patients with colon cancer (e.g. circRNA13452 and hsa_circ_0087391)? What are the results?

Response: We thank the reviewer for the suggestion. Actually, we have compared the predicting performance of single circRNA with four-circRNA-based risk score (cirScore) and presented the results in Fig 2E. To address your concerns, we have also built the Cox regression model using the combination of the two or three circRNAs and conducted the time-dependent AUC analysis. As shown, we observed that the predicting performance of the four-circRNA-based risk score (cirScore)

mostly outperforms than single circRNA or other circRNA combination in training cohort. The results are shown in revised Fig 2E and below for your review, and described in the revised manuscript (Page 8, Paragraph 1).

Figure R1. The left plot shown the time-dependent AUC of the two-marker combination signature and the cirScore, while the right plot depicted the value of the tree-marker combination signature and cirScore.

2) In page 9 and page 10, the number of tested nude mice should be 8 per group.

Response: Thanks for pointing out this issue! We have corrected the mistakes in our revised manuscript (Page 20, Paragraph 2).

3) As the result of knockdown of circRNAs (Figure S6B) is important, I suggest authors to move this data to Figure 6.

Response: We thank the reviewer for the suggestion. We have moved the panel of “knockdown of circRNAs” as Figs 4B and EV3C.

2nd Editorial Decision

14 May 2019

Thank you for the submission of your revised manuscript to EMBO Molecular Medicine. We have now received the enclosed report from the referee who was asked to re-assess it. As you will see this reviewer remains unsatisfied and still requires that two points previously raised be absolutely addressed. As these two points would necessitate extra-work and time, I have asked for editorial advice on this matter. Our external expert advisor fully agrees with the referee and stated the following: "I absolutely agree with this referee. The authors need to:

- 1- do the RNaseR resistant assessment (either by using Northern Blot or by adding a spike in before the RT qPCR)
- 2- do RNaseq to demonstrate that there are no off-target effects

So in my view the paper is not acceptable until these crucial experiments are done. Indeed the results of the experiments might preclude acceptance".

While we normally only offer one main round of revisions, in this case we have decided to make an exception. I would therefore strongly encourage you to address these two points experimentally as suggested. I'd like to say that depending on the nature of the revision, the paper may be sent back to referee #1.

Please submit your revised manuscript as soon as you possibly can.

***** Reviewer's comments *****

Referee #1 (Remarks for Author):

In the revision of the manuscript entitled "A circRNA signature predicts postoperative recurrence in stage II/III colon cancer" the authors addressed most of my concerns. In particular, the new experiments provide further support to the shRNA experiments. However, I am still not understanding why the authors can't perform a correct RNaseR resistant assessment (either by using Northern Blot or by adding a spike in before the RT qPCR). Only this will demonstrate that the effect is due to a circRNA and not to trans-splicing products. I am particularly worried about this as even w/o this normalization the circRNAs seem to be sensitive to RNaseR strongly suggesting that some of the detected molecules are not indeed circular. In addition, I am not convinced by the off-target assessment. Can't the authors perform RNAseq to rule this out?

2nd Revision - authors' response

28 June 2019

Referee #1 (Remarks for Author):

In the revision of the manuscript entitled "A circRNA signature predicts postoperative recurrence in stage II/III colon cancer" the authors addressed most of my concerns. In particular, the new experiments provide further support to the shRNA experiments.

Response: Thanks for your critiques and suggestions. We apologize for not providing convincing data for these two previously raised points. Here, we have addressed these two points experimentally as suggested.

1) I am still not understanding why the authors can't perform a correct RNaseR resistant assessment (either by using Northern Blot or by adding a spike in before the RT qPCR). Only this will demonstrate that the effect is due to a circRNA and not to trans-splicing products. I am particularly worried about this as even w/o this normalization the circRNAs seem to be sensitive to RNaseR strongly suggesting that some of the detected molecules are not indeed circular.

Response: We apologize for not providing convincing data for this point. The reason why we didn't perform a correct RNaseR resistant assessment is because we misunderstood "spike in" last time. Thus, the expression of β -Actin in mock-treated group was used for normalization in our qRT-PCR experiment with reference to other reports (Cell Research. 2015, 25:981-984; Mol Cancer. 2019 Feb 4;18(1):20; Nat Struct Mol Biol. 2015 Mar;22(3):256-64.). Now, we know that spike-in control is an exogenous (or external) control. We have modified our experiment procedure with reference to the other reports (Mol Cell. 2017 Apr 6;66(1):9-21.e7; Methods Mol Biol. 2016;1402:215-227.), and described as following:

RNaseR treatment of total RNA. Total RNA (4 μ g) was isolated with TRIzol reagent (Cat. # 15596-08, Life Technologies, Carlsbad, USA), and incubated with 3U/ μ g of RNase R (Epicentre Technologies, Madison, WI, USA) for 15 min at 37 °C or mock treated. The RNA was immediately transferred to ice, spiked with 10% mouse RNA, and extracted with TRIzol Reagent. The RNA concentration of the control group was determined, the same volume (6 μ L) of control RNA and the RNaseR treated RNA was used for reverse transcription. The circRNA expression levels and corresponding host gene' mRNAs were analyzed by qRT-PCR quantification, and the mouse GAPDH mRNA was used for normalization as an exogenous control.

The results indicated that these circRNAs were more resistance to digestion with RNase R exonuclease compared with linear host genes, which further confirmed that these circRNAs harbors a circular RNA structure (Fig EV2B). The above modified experiment procedure has been added in Appendix Supplementary Methods (Characterization of the selected circRNAs), the results shown below have been updated in Figure EV2B and described in our revised manuscript (Paragraph 1, Page 8).

Figure EV2B. qRT-PCR analysis for the expression of four selected circRNAs and the corresponding host genes after treatment with RNase R in HCT116 cells. The data was normalized to mouse GAPDH mRNA by adding a mouse RNA spike to each fraction. ** $P < 0.01$, Student's *t*-test.

2) In addition, I am not convinced by the off-target assessment. Can't the authors perform RNAseq to rule this out?

Response: We also apologize for not providing convincing data for this point. As suggested, we performed RNAseq and bioinformatic analysis to rule out off-target effects for the off-target assessment. The total RNA were prepared and subjected to RNA-sequencing study by the Corporation (Novogene, Beijing, China). The results demonstrated that knockdown of these circRNAs (hsa_circ_0122319, hsa_circ_0079480 and hsa_circ_0087391) had a high similarity of gene expression profile between the two independent shRNA group in SW620 and HCT116 cells (Spearman $R > 0.98$, $P < 0.001$, Fig EV3E), suggesting that the functional effects results from knockdown of these circRNAs rather than off targets.

For bioinformatics analysis, raw sequencing reads were processed with a prebuilt RNA-sequencing pipeline which could be free accessed at <https://github.com/likelet/RNAseqPipe>. Briefly, we first applied quality control step on the raw reads with fastp program. Then, the STAR and RSEM were adopted to align the reads against the hg38 reference genome and quantification in gene level, respectively. We then compared the TPM value of all known genes from different shRNA library and perform a correlation analysis with spearman's coefficient.

The RNA sequencing data of these cell lines are available in The Sequence Read Archive (SRA) (<https://www.ncbi.nlm.nih.gov/bioproject>), and accession ID is PRJNA551560. These results shown below has been added in Figure EV3E and described in the revised manuscript (Paragraph 1, Page 10; Paragraph 1, Page 20).

Figure EV3E. Correlation analysis of transcriptome between two independent shRNA groups in SW620 and HCT116 cells. In each scatter plot, the log₁₀ transformed Transcripts Per Million reads (TPM) of each gene were utilized for calculating the spearman's coefficient. R represent spearman's correlation coefficients, and P values were calculated by the spearman's correlation test.

3rd Editorial Decision

7 August 2019

Thank you for the submission of your revised manuscript to EMBO Molecular Medicine. We have now received the enclosed report from the referee who was asked to re-assess it. As you will see the reviewer is now supportive and I am pleased to inform you that we will be able to accept your manuscript pending the following final editorial amendments:

***** Reviewer's comments *****

Referee #1 (Remarks for Author):

The authors have addressed my concerns and the paper is now good for publication.

3rd Revision - authors' response

15 August 2019

The authors performed all minor editorial changes.

Corresponding Author Name: Rui-Hua Xu

Manuscript Number: EMM-2018-10168